# An Investigation of the Side Effects, Patient Feedback, and Physiological Changes Associated with Direct-Acting Antiviral Therapy for Hepatitis C

**DOI:** 10.3390/ijerph16244981

**Published:** 2019-12-07

**Authors:** Pin-Sheng Wu, Te-Sheng Chang, Sheng-Nan Lu, Hsiang-Jou Su, Shu-Zhi Chang, Chia-Wen Hsu, Mei-Yen Chen

**Affiliations:** 1Formosa Plastics Group Health Care, Yunlin 638, Taiwan; a0980192660@gmail.com (P.-S.W.); cristain854@gmail.com (S.-Z.C.); 2Division of Gastroenterology and Hepatology, Department of Internal Medicine, Chang Gung Memorial Hospital, Chiayi 613, Taiwan; cgmh3621@cgmh.org.tw (T.-S.C.); juten@ms17.hinet.net (S.-N.L.); 3College of Medicine, Chang Gung University, Taoyuan 333, Taiwan; 4Department of Nursing, Chang Gung University of Science and Technology, Chiayi 613, Taiwan; s051037@mail.cgust.edu.tw; 5Formosa Plastics Group, Safety Health & Environment Center, Taipei 508, Taiwan; kevinhsu2@fpg.com.tw; 6College of Nursing, Chang Gung University of Science and Technology, Chiayi 613, Taiwan; 7Department of Cardiology, Chang Gung Memorial Hospital, Chiayi 613, Taiwan; 8School of Nursing, Chang Gung University, Taoyuan 333, Taiwan

**Keywords:** Hepatitis C virus, direct-acting antivirals (DAAs), sustained virologic response (SVR), rural

## Abstract

***Background:*** Hepatitis C virus (HCV) infection is one of the major causes of liver cirrhosis and hepatocellular carcinoma globally. The advent of direct-acting antivirals (DAAs) with high cure rates provides an opportunity to reduce the rising HCV disease burden. However, few studies have explored the side effects and physiological benefits of DAA therapy in rural areas. The aim of this study was to investigate the subjective reports of discomfort, patient feedback about the course of treatment, and physiological changes after DAA treatment in HCV patients. ***Methods:*** A descriptive, prospective, comparative cohort study was conducted from January to August 2019 in western coastal Yunlin County, Taiwan. Data regarding demographic characteristics, subjective discomfort levels, and physiological responses were collected through face to face interviews and from medical records by a cooperating hospital. ***Results:*** Six-hundred-and-twenty-three participants with an active HCV infection were identified; 555 (89.1%) had completed treatment, and sustained virologic response was achieved in 99.6% (n = 553). The mean age was 64.9 (standard deviation = 13.1) years, and 35% of patients experienced discomfort during DAA treatment, including fatigue, itching, and dizziness. After three months of treatment, physiological markers, including body weight (*p* < 0.001), waist circumference (*p* < 0.05), blood pressure (*p* < 0.001), alanine aminotransferase (*p* < 0.001), and aspartate aminotransferase (*p* < 0.001), had significantly improved. Almost all participants provided positive feedback about the treatment experience and reported manageable side effects. ***Conclusions:*** The findings showed that, in an endemic rural area, DAA treatment had a high cure rate and improved physiological markers with few discomforts. These results can be used to reduce the barriers HCV patients face in adopting new medications.

## 1. Introduction

Chronic hepatitis C (CHC) virus infection is a major cause of liver cancer and cirrhosis. In addition, some studies have indicated that patients with CHC have both higher prevalence of type 2 diabetes and increased cardiovascular risk compared to those who have never been infected [1,2]. Globally, an estimated 71 million people have CHC, which is caused by the hepatitis C virus (HCV), a blood borne virus. The most common modes of infection are through exposure to small quantities of blood, such as through injection drug use, inadequate sterilization of medical equipment, or transfusion of unscreened blood and blood products [3,4]. Globally, there is a considerable burden of CHC and HIV infections among people who inject drugs, and the transmission of both infections continues [5,6]. Unlike the transmission modes described above, the majority of CHC patients in Taiwan are elderly, born in and around the 1950s, and are likely to have been infected by unsafe medical devices, especially in rural areas [3]. The current prevalence of HCV infection in Taiwan is about 4% [3].

The treatment of HCV infection has evolved from interferon-based treatment approaches to direct-acting antivirals (DAAs). Recently, the emergence of DAAs, such as Asunaprevir/daclatasvir, Elbasvir/Grazoprevir, and Ledipasvir/Sofosbuvir, has increased cure rates and treatment safety, with more than 95% efficiency, few adverse effects, and a short treatment time of 8–12 weeks [7,8,9,10]. The World Health Organization has developed a global health strategy to eliminate hepatitis C by 2030 [4], and the Taiwanese government also launched an elimination goal by 2025 [3]. In addition, previous literature confirmed that DAA treatment is not only beneficial for HCV clearance but is also associated with better fasting glucose and controlled glycated hemoglobin levels and improved cardiovascular function [2,11].

Although the advent of free DAA treatment provides an opportunity to reduce the rising burden of this disease, many countries continue to face barriers to diagnosis and treatment, especially in resource-limited settings [6,12]. Additionally, patients who have been identified as HCV carriers are not properly educated or made aware of the need for medication, or they feel restricted by the requirement to adhere to medication and choose not to do so [13,14,15]. Therefore, reducing the global burden of HCV infection must depend upon the success of preventive interventions as well as the implementation of outreach screening [3,4]. Primary healthcare providers can play an important role in advising potential HCV patients to seek a confirmed diagnosis and free DAA treatment, which should be supplemented with individualized, culturally-tailored care until the completion of the treatment course. However, information related to the side effects of DAA therapy and feedback regarding the treatment experiences of HCV patients are lacking. Therefore, the purpose of this study was to explore the subjective reports of discomfort and treatment experience as well as physiological markers after HCV treatment. 

## 2. Materials and Methods

### 2.1. Design, Sample, and Setting

This report was one of a series of prospective cohort studies, which were conducted by a nurse-led community health development program and supported by a government grant for reducing HCV infection burden around the western coastal rural areas of Taiwan from August 2018 to July 2021. A descriptive, prospective, comparative cohort study was conducted. Participants who were found to be positive for anti-HCV antibodies in the medical records were invited to the collaborating hospital for HCV RNA confirmation and free DAA treatment by the research team. 

### 2.2. Procedure and Ethical Consideration

This study was approved by the institutional review board (IRB No. 201701919B0). A total of 1795 patients had tested positive for the anti-HCV antibody; of these, the research team transferred 623 adults with HCV RNA detected in their blood to a local hospital for further treatment. The research team conducted face to face interviews at outpatient clinics to collect data regarding side effects and feedback related to the free DAA treatments. These interviews were conducted upon completion of the treatment course (2–3 months), and again, 3 months after the end of the treatment.

### 2.3. Measurements

#### 2.3.1. Demographics and Basic Health-Related Behaviors 

We collected demographic data including age, sex, education level, serum HCV genotype, and current or past use of DAAs from the medical records. Participants were asked if they (a) were aware of free medication therapy; (b) had been diagnosed with diabetes mellitus, hypertension, or heart disease by a doctor; (c) had a family member infected by HCV; or (d) had been treated with antivirals in the past. The participants’ health-related behaviors pertaining to liver health were recorded according to previous literature [16,17]. We included six health-related habits, and responses were categorized as frequent for usually/always and infrequent for never/seldom. Participants were asked (a) how often they followed a healthy diet that included the 5 food groups (e.g., rice, meat, vegetables, fruit, and milk); (b) how often they drank at least 1500 mL water; (c) whether they engaged in regular exercise for 30 minutes/day, 3 times each week; and (d) whether they smoked cigarettes, chewed betel nuts, and drank alcohol (currently, never, or formerly).

#### 2.3.2. The Subjective Discomfort (Side Effects)

We asked participants’ subjective feelings about adopting the DAA treatment and recorded these answers by telephone interview every week. We asked, “Do you have any discomfort this week?” 

#### 2.3.3. Physiological Biomarkers 

Physiological data were collected from patient medical records by the cooperating hospital and included (1) body weight, (2) waist circumference (cm), (3) systolic/diastolic blood pressure (mmHg), (4) alanine aminotransferase (ALT) and aspartate aminotransferase (AST) levels, (5) HCV RNA confirmation testing used to classify participants as negative (those with viral clearance or sustained virologic response (SVR)) or positive.

#### 2.3.4. Feedback Related to the Free DAA Treatments 

We asked participants’ for feedback regarding their experience after completion of the 2–3-month treatment course. We used the following questions in face to face interviews: (a) immediately after treatment completion, we asked, “How do you think you have finished the treatment course?” and (b) at 3 months after completion of treatment we asked, “How do you think that you have cleaned out the virus (germ/worm)?”

### 2.4. Statistical Analyses

SPSS version 22 (IBM SPSS Inc, Chicago, Illinois) was used for data analyses. A descriptive and dependent sample t-test was used to assess the subjective reports of the side effects or responses to the treatment-related questions, as well as the physiological biomarkers (continuous variables). All tests were two-tailed, and *p* < 0.05 was considered statistically significant. 

## 3. Results

We screened 623 participants with active HCV infection, 68 of whom refused to transfer for further DAA treatment. Ultimately, 555 completed the treatment (89.1%), and 553 (99.6%) tested positive for HCV RNA clearance. The mean age of the participants was 64.9 (standard deviation = 13.1, range 20–92) years, with more than half of them aged >65 years. Most participants reported having received little education (85%) and having hypertension (38.6%), diabetes (18.2%), and heart diseases (12.6%), which had been diagnosed by a doctor (Table 1).

The major genotypes of HCV were type 1b (n = 336, 60.5%) and type 2 (n = 191, 34.4%). More than one quarter (n = 159, 28.7%) of the participants reported having a family member with HCV infection, and 13.2% reported having been treated with interferon-based medications without success. More than two thirds (78%) had been treated for 12 weeks, and the top three medications used were Elbasvir/Grazoprevir (46.4%), Ledipasvir/Sofosbuvir (29.9%), and Glecaprevir/Pibrentasvir (23.4%). More than half (62.1%) were classified as overweight or obese. Nearly one third (35%) reported discomfort (side effects) and 99.6% (n = 553) showed virus clearance at three months after completing treatment (Table 1). Regarding health-related behaviors before starting DAA therapy, 43.5%, 50.1%, and 79.1% reported that they infrequently followed a healthy diet, consumed adequate water, or exercised regularly, respectively. In addition, many participants still consumed alcohol, betel nuts, and cigarettes.

As shown in Table 2, the top three discomfort symptoms reported included fatigue (sleepiness and drowsiness, 70.1%) and skin issues (e.g., itching, 40.7%), and more than one-third experienced headache, dizziness, or gastroenterology-related discomfort. Table 3 shows the physiological marker levels before and after receiving DAA therapy. The paired *t*-test showed that body weight (*t* = 4.82, 95% confidence interval (CI) = 0.34–0.80, *p* < 0.001), waist circumference (*t* = 2.09, 95% CI = 0.01–0.21, *p* < 0.05), systolic blood pressure (*t* = 4.61, 95% CI = 2.26–5.61, *p* < 0.001), diastolic blood pressure (*t* = 3.57, 95% CI = 1.91–3.13, *p* < 0.001), ALT (*t* = 13.48, 95% CI = 29.47–39.53, *p* < 0.001), and AST (*t* = 12.43, 95% CI = 16.38–22.25, *p* < 0.001) showed significant improvements after therapy. 

Table 4 shows that many participants responded positively regarding their experience immediately following the treatment course and also three months after DAA treatment. For instance, immediately following treatment, some participant responses included, “Thanks for your constant phone calls and comfort to accompany my treatment completion… It seemed like a marathon race… Without your reminders and encouragement, I would have already given up… How could I ever have completed the treatment?” Three months after the treatment course, many participants were confirmed as being virus-free, and most of them responded with phrases like, “I feel much happier and more excited than if I had won the lottery bonus…I can live a longer, healthier life… You really didn’t lie to me… The drug is very effective… The medicine is powerful… Thanks to your team…described by the family members.”

## 4. Discussion

To the best of our knowledge, few studies have reported the levels of discomfort, side effects, and physiological changes experienced by patients during and after DAA treatment. Four key findings emerged from this study. First, the nurse-led community health development program significantly benefitted rural adults with HCV by advocating medication adherence and treatment completion, and these patients had high cure rates. Second, participants exhibited greater improvement in the physiological markers, such as body weight, waist circumference, blood pressure, and liver function. Third, nearly one-third of participants reported discomforts; however, most of them were manageable. Lastly, the positive feedback related to free DAA therapy strongly supported the government’s free DAA treatment policy. 

The present study showed that most of the participants were older, female, of low social economic status, had undergone 12 weeks of treatment, and were predominantly infected with the viral genotypes 1b and 2. With the exception of the age and sex of our cohort, much of the demographic characteristics were similar to those of cohorts studied in Japan and Australia [13,18,19]. For instance, in Australia, White et al. found that the median age of patients was 48 years, 66% of participants were male, 26% had been previously treated with eight-week regimens, and 57% tested positive with genotype 1 [19]. In South Korea, HCV patients had a mean age of 57.3 years, and the major genotypes detected were 1b (48.2%) and 2 (46.4%) [20]. Cohort differences might be due to the unsafe medical device transmission route in our study in rural areas during the baby boomer era, i.e., from 1945–1965 [21,22]. 

Surprisingly, there were four participants who were more than 80 years old in the present study who also showed virus clearance following DAA therapy. This is in agreement with a report from the United States by Pan et al. [21], who evaluated HCV SVR rates and showed that DAAs were associated with high rates of SVR in all age groups. The age of patients does not seem to have a significant impact on the efficacy of DAAs, even when patients are in the oldest age category (≥75 years). Therefore, DAA treatment should not be withheld from older individuals [21]. More than one quarter of participants in our study had family clustered infections; 28.7% had more than one relative infected by HCV. Further studies could adopt a screening program that includes all family members to accurately identify all patients with HCV.

The rate of SVR achieved in our study was higher than that in a study in Australia by White et al., who enrolled 327 patients who underwent assessment and commenced treatment in primary care settings with an SVR rate of 95.6% [19]. Moreover, the medication adherence was also higher in this study than that in a study conducted in a psychiatry department in Sweden by Sundberg et al., who showed that adherence to DAA treatment was estimated at 95%, and the SVR rate was 88% [23]. The high medication adherence and SVR rate in our study might be attributable to the individualized care provided by the research team with frequent telephone calls to ensure medication adherence; moreover, we also conducted prompt consultations.

In the study, 73 patients, who had been treated with interferon-based medications in vain, still were invited by our insistent phone care based on the following motivations. (1) They were encouraged by the successful DAA treatment cases from their own neighbors or families. (2) It is much more convenient for only oral medication daily at home than weekly injection by nurses in hospital. (3) DAA treatment only takes 8–12 weeks; it’s easier than interferon-based medications 24–48 weeks tolerance. (4) The VIP services by the research team, such as hospital registration, outpatient accompany, weekly phone care, and reminding, even medical transportation arrangements. In addition, from the above 73 patients’ previous painful interferon-based medications experiences, most felt headache, dizzy, exhausted, fever, hair loss, itching, poor appetite, dysphagia, nausea, vomit, stomach ache, diarrhea, constipation, weight loss, night sweat, depressive mood, insomnia, anxiety, oral ulcer, and dyspnea in activities, etc. Nevertheless, the phenomenon of discomfort by DAA treatment only was shown in Table 2. In Table 4, participants’ positive feedbacks during different phases exactly reflected the above patients’ happy feelings. 

The present study showed an improvement in physiological biomarkers at the end of treatment. This result is consistent with previous study findings that DAAs not only eliminate HCV but also significantly improve both systolic/diastolic blood pressure and liver function [1,11]. HCV was found to be a risk factor for cardiometabolic disease, vascular coronary artery disease, and type 2 diabetes, which are extra-hepatic manifestations [1,11]. Many CHC patients in the present study exhibited hypertension and diabetes (Table 1). Clinicians and primary healthcare providers can use these results to invite CHC patients to undergo free DAA treatment.

Many CHC patients reported the lack of a healthy diet and not drinking adequate water and exercising. Although an assessment of these health-related behaviors was not the purpose of this study, an unhealthy lifestyle may be correlated with some cardiometabolic diseases [16]. Further studies should explore these health-related habits among CHC patients. Although nearly one-third of the participants reported experiencing side effects during the two–three months of treatment, most of these effects were manageable. Moreover, almost all participants provided positive feedback regarding their experience of undergoing free DAA therapy and strongly supported the government’s policy.

## 5. Conclusions

Altogether, in HCV endemic rural areas, short duration DAA treatment significantly influenced the medication adherence and treatment completion rates among HCV carriers, which, in turn, led to a high cure rate and the improvement of physiological markers with few discomforts. Although 35% of HCV carriers in this study reported experiencing side effects, the top three being fatigue, itching, and dizziness, all these side effects were considered manageable, and most participants provided positive feedback after the end of treatment. These results can be used to reduce the barriers HCV patients face in adopting new medications.

## Figures and Tables

**Table 1 ijerph-16-04981-t001:** Demographic characteristics and health-related behaviors (N = 555).

Variables	N (%)		
Gender		Have heard DAAs treatment	
Male	258 (46.5)	Yes	238 (42.9)
Female	297(53.5)	No	317 (57.1)
Age (years)	Mean = 64.9; SD = 13.1; Range 20~87		
<65	240 (43.2)	Chronic diseases	
≧65	315 (56.8)	Hypertension	214 (38.6)
Education level (years)		Diabetes	101 (18.2)
Illiterate	237 (42.7)	Heart disease	70 (12.6)
≦ 9	235 (42.3)	Family clustered (person)	
≧ 12	83 (15.0)	0	319 (57.5)
HCV genotype		1	135 (24.3)
1a	12 ( 2.2)	2–6	24 ( 4.4)
1b	336 (60.5)	Missing	77 (13.8)
2	191 (34.4)	Have treated by interferon-based medications	
3	4 ( 0.7)	Yes	73 (13.2)
6	12 ( 2.2)	No	482 (86.8)
Treatment course		Discomfort (side effects)	
8 weeks	122 (22.0)	Yes	194 (35.0)
12 weeks	433 (78.0)	No	361 (65.0)
Received DAAs		HCV (RNA)	
Zepatier	258 (46.5)	Negative (clearance)	553 (99.6)
Harvoni	166 (29.9)	Positive	2 ( 0.4)
Maviret	130 (23.4)	Alcohol drinking	
Sofosbuvir	1 ( 0.2)	Never/former	499 (89.9)
Body mass index		Current	56 (10.1)
Normal (~24)	210 (37.8)	Betel-nut chewing	
Overweight (24~27)	170 (30.6)	Never/former	460 (82.9)
Obesity (27.1~)	175 (31.5)	Current	95 (17.1)
Adopt healthy diet (5 groups)		Cigarette smoking	
Frequent	314 (56.5)	Never/former	443 (78.0)
Infrequent	241 (43.5)	Current	122 (22.0)
Drinking water (>1500 mL/day)		Regular exercise (30 min/day)	
Frequent	277 (49.9)	Frequent	116 (20.9)
Infrequent	278 (50.1)	Infrequent	439 (79.1)

RNA, ribonucleic acid; DAA, direct-acting antiviral.

**Table 2 ijerph-16-04981-t002:** The phenomenon of discomfort (N = 194).

Classification	Contents of description	N (%)
1. Fatigue	Fatigued, sleepiness, drowsiness	136 (70.1)
2. Skin	Itching, urticaria, losing hair	79 (40.7)
3. Head	Dizzy, headache	74 (38.1)
4. Gastroenterology	Dry mouth, stomachache, nausea, vomiting, constipation, diarrhea, no appetite, heartburn	73 (37.6)
5. Others	Palpitation, blurred vision, night sweats, insomnia, chest pain	10 (5.2)

**Table 3 ijerph-16-04981-t003:** Biomarkers changed before and after direct-acting antiviral (DAA) treatment (N = 555).

Variables	Mean (SD)		*t*	*p*	95% CI *
	Before	After			
Body weight	64.44 (12.63)	63.87 (12.72)	4.82	<0.001	0.34–0.80
WC (N = 537)	86.10 (11.46)	85.99 (11.42)	2.09	0.038	0.01–0.21
SBP	136.31 (20.39)	132.47 (19.99)	4.61	<0.001	2.26–5.61
DBP	77.76 (13.38)	75.74 (12.77)	3.57	<0.001	1.91–3.13
ALT	57.49 (56.5)	22.99 (39.5)	13.48	<0.001	29.47–39.53
AST	43.28 (34.01)	23.97 (20.94)	12.93	<0.001	16.38–22.25
Bilirubin (Total)	0.62 (0.33)	0.64 (0.34)	−1.97	0.050	−0.05 to −0.01
Bilirubin (Direct)	0.21 (0.14)	0.22 (0.19)	−1.13	0.260	−0.03 to −0.01

* Confidence interval; WC, waist circumference; SBP, systolic blood pressure; DBP, diastolic blood pressure; ALT, Alanine aminotransferase; AST, Aspartate aminotransferase.

**Table 4 ijerph-16-04981-t004:** The participants’ positive feedbacks during different phases.

**Phase 1—Completion of DAA treatment (the end of treatment)**
Thanks for your constant phone calls and comforts to accompany my treatment completion. It looks like a marathon racing…crying… Without your reminders and encouragement, I have already given up, how could I complete the treatment? I love you… I finally reached the destination of hard treatment with your team’s full supports. Since no more DAA medicine, I should be able to sleep well and sound, right? Will the side effects disappear automatically after the treatment? Finally, I don’t have to worry about forgetting to take medicine on time every day. After treatment completion, will there still be any your team to accompany me to the clinic and chat together? Three months have passed quickly; it’s great that my treatment is eventually finished!
**Phase 2—Virus confirmation (3 months after the end of treatment)**
I will introduce my father-in-law and my relatives to take DAA. I feel much happier and more exciting than the lottery bonus. Fortunately, the hard work has not been in vain. I can live healthily much longer. Are the worms in the stomach annihilated? You really didn’t lie to me, the drug is very effective. My ancestors really bless me. The modern medicine is very powerful, and soon the cancer will not be a threat any more. I can live longer…these drugs are so powerful… crying…I love you all… Thank you for your team …assist my parents to complete treatment…from family members described.

DAA, direct-acting antiviral.

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
