# Peer review of "An Investigation of the Side Effects, Patient Feedback, and Physiological Changes Associated with Direct-Acting Antiviral Therapy for Hepatitis C"

_ijerph, 2019, doi:10.3390/ijerph16244981_

Round 1

Reviewer 1 Report

Congratulations for your paper. The use of DAAs for treating chronic HCV infection is still a controversy for patients in the sense of side effects ocurrence and negative impacts of interferon-based therapies. It was satisfatory to describe that in HCV endemic rural areas, DAA treatment has shown high SVR rates because of a good medication adherence. 

I was wondering about a better description for 73 patients who reported having been treated with interferon-based medications without success. Since they had a negative result with a previous treatment, how was their motivation for DAA treatment? The occurrence of previously side effects during interferon therapy could have been a limitation to start a new treatment with DAAs? My interest is about a comparative feedback between previous and present therapy. If possible, I suggest a better discussion for these patients.

Author Response

Manuscript ID:ijerph-657295

Many thanks for your kindly and important comments. We have reedited and corrected the manuscript for reviewer’s comments with red font. We have followed the comments and reedited by a native English speaker.

For reviewer 1:

Regarding bout a better description for 73 patients who reported having been treated with interferon-based medications without success. Since they had a negative result with a previous treatment, how was their motivation for DAA treatment? The occurrence of previously side effects during interferon therapy could have been a limitation to start a new treatment with DAAs? My interest is about a comparative feedback between previous and present therapy. If possible, I suggest a better discussion for these patients.

Ans. Thank you for your comments, we have provided more information on p7 with red font.

Reviewer 2 Report

This is a report on the side effects, patient feed back and physiological changes after DAA treatment. The results may provide a kind of informative results, however, there are several flaws should be addressed.

Comments

Abstract: line 27: please check the period of study. Are you sure to be performed from January to December 2019 (which is not completed)? Introduction: It is better to provide data on the current prevalence of HCV infection in Taiwan. Line 58-59. It might be better to mention generic name of DAAs instead of mentioning trade name only. Materials and Methods, page 3, line 118: “month” should be “months”. Results, Table 1 Sofosbuvir is generic name and other 3 DAA is trade name. Also No. of never cigarette smoking should be 433 instead of 443. Results, Table 3 : Title should be start with capital and remove a comma. “ Bilirunin “ should be “Bilirubin”. Discussion, page 6, line 183 : “demographic characteristics were similar to those of cohorts studied in other countries”. It is better to specify which countries have been used for the comparison. Discussion, page 6, lines 207-209: The reference of the previous findings is needed. 

Author Response

For reviewer 2:

Regarding on the Abstract: line 27: please check the period of study.

Ans. Thanks for your patient and precise suggestions, the exact performed period should be from January to August 2019, we have reedited on p1 with red font.

Regarding on the Introduction: It is better to provide data on the current prevalence of HCV infection in Taiwan.

Ans. We have reedited on p2 with red font.

It might be better to mention generic name of DAAs instead of mentioning trade name only.

Ans. We have reedited on p2 with red font.

Materials and Methods, page 3, “month” should be “months”.

Ans. Thank you. We have corrected “months” instead of “month” on p2 with red font.

Results, Table 1 Sofosbuvir is generic name and other 3 DAA is trade name.

Ans. We have corrected the generic name on p4 and table 1 with red font.

Regarding the No. of never cigarette smoking should be 433 instead of 443. Results, Table 3 : Title should be start with capital and remove a comma. “Bilirunin“ should be “Bilirubin”.

Ans. Thank you again. We have reedited table 1 and 3 with red font.

Discussion, page 6, “demographic characteristics were similar to those of cohorts studied in other countries”. It is better to specify which countries have been used for the comparison.

Ans. We have reedited on p6 with red font.

Discussion, page 6, The reference of the previous findings is needed.

Ans. We have reedited on p6 with red font.